# Advances in Plant-Derived Extracellular Vesicle Extraction Methods and Pharmacological Effects

**DOI:** 10.3390/biology14040377

**Published:** 2025-04-06

**Authors:** Nuerbiye Nueraihemaiti, Dilihuma Dilimulati, Alhar Baishan, Sendaer Hailati, Nulibiya Maihemuti, Alifeiye Aikebaier, Yipaerguli Paerhati, Wenting Zhou

**Affiliations:** 1Department of Pharmacology, School of Pharmacy, Xinjiang Medical University, Urumqi 830017, China; nurbiye@stu.xjmu.edu.cn (N.N.); dilihuma@stu.xjmu.edu.cn (D.D.); alhar@stu.xjmu.edu.cn (A.B.); sendaer@stu.xjmu.edu.cn (S.H.); nurbiye24@stu.xjmu.edu.cn (N.M.); alifeiye@stu.xjmu.edu.cn (A.A.); yipaerguli@stu.xjmu.edu.cn (Y.P.); 2Xinjiang Key Laboratory of Natural Medicines Active Components and Drug Release Technology, Urumqi 830017, China; 3Xinjiang Key Laboratory of Biopharmaceuticals and Medical Devices, Urumqi 830017, China; 4Engineering Research Center of Xinjiang and Central Asian Medicine Resources, Ministry of Education, Urumqi 830017, China

**Keywords:** plant-derived extracellular vesicles, anti-inflammatory, anticancer, anti-infective, extraction

## Abstract

Plant-derived extracellular vesicles (PDEVs) are promising as non-invasive biomarkers and drug delivery carriers in studying and treating complex diseases, like inflammatory diseases, infectious diseases, and tumors. Secreted mainly by plants, PDEVs outperform human cell-derived extracellular vesicles in safety, stability, and biocompatibility, serving as valuable tools for exploring disease mechanisms and therapeutic targets. However, crucial issues in this field demand immediate attention. Existing extraction techniques are inadequate and inefficient, failing to meet the needs of large-scale, high-quality extraction. Regarding pharmacological effects, there is a lack of thorough understanding of PDEVs’ in vivo mechanisms, efficacy, and potential side effects. These knowledge and technological gaps impede the clinical application and promotion of PDEVs. Therefore, resolving these issues is vital for advancing related fields and refining diagnostic and treatment strategies. This article will summarize and analyze PDEVs’ extraction methods and explore their latest pharmacological progress, offering references to researchers and facilitating further PDEV research and application.

## 1. Introduction

Extracellular vesicles (EVs) are membranous vesicles that are released by cells, including exosomes, microvesicles, and apoptotic bodies [1]. Exosomes (Exos) are membrane vesicles that are released by all cells into the surrounding environment with a particle size of 40–200 nm [2]. Microvesicles are membrane vesicles that are released from the plasma membrane into the extracellular environment by outgrowth and fission in the form of vesicles with an approximate diameter of 100–1000 nm [3], which were first reported by Wolf (1967) as plasma released from intact platelets that contain tiny particles that can be extracted by ultracentrifugation [4]. Apoptotic vesicles (ABs) are membrane vesicles with a diameter of 1–5 μm and are a major subset of EVs that are released by apoptotic cells, which are mainly formed by apoptotic cells through catabolic processes, including plasma membrane vesiculation, apoptotic membrane protrusion formation, and fragmentation [5]. Almost all cell types can secrete EVs into the extracellular environment, so EVs are widely present in blood [6], urine [7], various tumor cells (cholangiocarcinoma cells, hepatocellular carcinoma cells, and colorectal cancer cells [8,9,10]), and stem cells. Their role in intercellular communication and interactions between different tissues is incredibly important. Studies have found EVs to be involved in cancer, inflammation, cardiovascular disease, lung disease, neurodegenerative diseases [11,12,13,14,15], and the pathological and physiological processes of diseases. EVs have been studied extensively in the last two decades, but research by scientists on EVs has focused mainly on the extraction and purification of EVs from cells or body fluids and their functional and morphological characterization. Relatively little research has been conducted on plant-derived extracellular vesicles (PDEVs), which mainly serve as protective compartments in plants for the intercellular transportation of various substances, contributing to plant growth and development, defense responses, and plant–microbe symbiosis. PDEVs are similar to those of mammals in that they can be classified into several subclasses based on their biogenesis pathway. Three subtypes of EVs have been isolated so far in Arabidopsis, including tetraspanin-positive exosomes derived from multivesicular vesicles (MVBs), permeabilized 1 (PEN1)-positive EVs, and EVs derived from extracellular vesicle-positive organelles (EXPOs) [16]. PDEVs also contain an abundance of bioactive substances, such as proteins, nucleic acids (DNA, mRNA, miRNA, lncRNA), and lipids [17]. Studies that have analyzed the protein composition of PDEVs have identified several key proteins in PDEVs, including membrane-bound proteins, heat shock proteins, aquaporins, actin, syntaxins, clathrin heavy chains, and RAS-related proteins [18]. The research found clathrin heavy chains, heat shock proteins, and 14-3-3 proteins to be highly expressed in microvesicles and nanovesicles isolated from four citrus species (C. sinensis, C. limon, C. paradisei, and C. aurantium), while aquaporins were only highly expressed in nanovesicles [19]. An analysis of PDEV bioactive substances also found the presence of miRNAs and the presence of microRNA (miRNA aly-miR396a-5p) in ginger-derived exosome-like nanoparticles (GELNs) that were demonstrated to be a therapeutic agent for inhibiting inflammation in the lungs [20]. In addition, therapeutic miRNAs, such as this one, have also been found in grapes, strawberries, and apples. Lipid analysis in PDEVs has also attracted a certain amount of interest, and the main lipid species found in PDEVs are phosphatidic acid (PA), phosphatidylethanolamine (PE), and phosphatidylcholine (PC) [21]. The lipidome analysis of EVs of tea origin (TLNT) showed 40% PC, 11% PE, and 7% PA [22], and PDEVs were found to play an important role in the treatment of inflammation [23], tumors [24], and other disease processes. It is of great significance to conduct in-depth research on PDEVs. On the one hand, our current understanding of PDEVs is far less than that of animal-derived extracellular vesicles. Many aspects, such as the differences in PDEVs among different plant species and their unique regulatory mechanisms, remain to be studied. On the other hand, considering the potential of PDEVs in disease treatment, in the future, efforts should be focused on optimizing the extraction and purification techniques of PDEVs, improving their yield and purity, and reducing costs to meet the needs of research and application. Therefore, in this paper, we have collected articles on PDEVs published over the past decade and sorted out the isolation methods, biological functions, and characterizations of medicinal and edible plants, such as grapefruit, strawberry, ginger, ginseng, tomato, cabbage, momordica charantia, lemon, orange, tea, broccoli, garlic, blueberry, carrot, panax notoginseng, *solanum nigrum* L., etc. (Table 1). This is aimed at gaining a deeper understanding of the synergistic mechanism of bioactive substances within PDEVs, exploring their applications in the treatment of more diseases, and providing strong support for overcoming complicated diseases and developing new treatment methods.

## 2. Characterization, Identification, and Isolation Techniques for Plant-Derived Extracellular Vesicles (PDEVs)

### 2.1. Characterization and Identification of PDEVs

The structural features of PDEVs are similar to animal exosomes as they are membrane structures with lipid bilayers, membrane proteins on the surface, and proteins, nucleic acids and other substances on the interior, although they have a particle size that is generally slightly larger than that of animal EVs [55]. The particle size is normally identified using nanoparticle tracking analysis (NTA), in addition to the size distribution, which is characterized and identified by transmission electron microscopy (TEM). The average particle size of EVs extracted from Arabidopsis leaves was found to be in the 90–100 nm range [16]. The average sizes of microvesicles (MV) and nanovesicles (NV) extracted from tomato were found to be in the range of 110 ± 10 nm and 155 ± 10 nm [28], and the separation of EVs from cabbage using PEG precipitation, ultracentrifugation, and size-exclusion chromatography found the particle sizes to be 148.2, 134.2, and 98.8 nm, respectively. NTA was used to analyze the morphological characteristics of cabbage, and they were found to have a spherical average size of 100 nm, an average zeta potential of −14.8 mV, and good stability [31]. Broccoli-derived EVs (BDEVs) are well-stabilized in the diameter range of 50–150 nm [50]. These nanovesicles generally have a small size range of between 30 nm and 1500 nm and a negative Zeta potential of above −20 mV, demonstrating high stability [56].

### 2.2. Separation Techniques for PDEVs

The efficient extraction of plant extracellular vesicles is a prerequisite for ensuring the stability and bioactivity of PDEVs. As interest in the study of plant extracellular vesicles has increased, several studies have been conducted to improve the specificity and efficiency of extraction, with each method having different advantages and disadvantages. Ensuring the intact bilayer structure of the PDEVs in the process of extraction and storage should be given attention to prevent the bioactivity from being affected. The first large-scale survey on EV extraction and separation in 2016 found that ultracentrifugation (including differential centrifugation) was used by 81% of the population, followed by density gradient centrifugation (20%), ultrafiltration (18%), and size-exclusion chromatography (SEC, 15%) [57].

#### 2.2.1. Ultracentrifugation (UC)

The differential ultracentrifugation method is the gold standard for extracting EVs and separates EVs by the buoyancy density of particles [58]. The centrifugation steps are as follows: the first step is to precipitate the main part of the cells using low-speed centrifugation (300–400× *g*) for 10 min, then 2000× *g* and 10,000× *g* are used sequentially to remove cellular debris and other structures with higher buoyancy density than the EVs. The final step is to use ultracentrifugation (100,000–200,000× *g*) to isolate the resulting supernatant from the contained EVs before repeated ultracentrifugation to remove non-vesicular proteins from EVs [59]. The resulting EVs are then further purified as a means of eliminating contamination. Purification and decontamination are achieved using washing and microfiltration. Washing involves placing the precipitate in a large amount of PBS and centrifuging it at 100,000× *g* (Figure 1). The disadvantage of this method is the long time it requires (4–5 h) and the relatively low recovery [60]. Microfiltration uses filters with pore sizes of 0.1, 0.22, or 0.45 μm to select the particles to be separated based on particle size. It should be noted that the microfiltration process increases EV purity while significantly reducing the yield of vesicles. The efficiency of ultracentrifugation for separating EVs depends on many factors, including the geometric parameters of the rotor (k-factor or scavenging factor), rotational speed, centrifugation duration, and the viscosity of the solution [61]. Reducing the viscosity of the fluid, lowering the scavenging factor or k-factor of the rotor, and reducing the centrifugation time can significantly improve the settling efficiency and a higher settling efficiency results in a higher recovery of EVs. The advantages of ultracentrifugation include its suitability for separating EVs from large amounts of biofluids and that it requires relatively few reagents and consumables [58], while its disadvantages are that it requires specific equipment (ultrahigh-speed centrifuges), that it is time-consuming and specialized, and that specific knowledge is required to be able to perform it [62] (Table 2).

#### 2.2.2. Density Gradient Ultracentrifugation (DGU)

In the density gradient ultracentrifugation technique, the separation of EVs is achieved in a preconstructed density gradient medium within a centrifuge tube, relying on the size, mass, and density of the EVs. The density of this density gradient medium gradually increases from top to bottom [63]. According to the differences in the buffers used, density gradient ultracentrifugation can be further subdivided into two types: sucrose-based density gradient centrifugation and iodixanol-based density gradient centrifugation. Some studies have shown that using 60% iodixanol density gradient centrifugation can maximize the recovery rate of EVs. It can not only maintain the physical integrity and biological activity of EVs but can also improve the separation purity of EVs [64]. The combined use of sucrose density gradient centrifugation and ultracentrifugation can significantly optimize the separation effect and effectively ensure the integrity of the extracellular vesicle membrane structure. Therefore, this technical combination has been widely applied in the extraction and separation of plant-derived EVs [44]. However, the density gradient ultracentrifugation technique also has some obvious limitations. On the one hand, the operation process is rather cumbersome, requiring a large amount of manpower, and the entire process is time-consuming, which often leads to a large loss of EVs. On the other hand, this technique has high requirements for equipment and requires the installation of expensive instruments, which restricts its wider popularization and application to a certain extent [65].

#### 2.2.3. Ultrafiltration (UF)

Ultrafiltration involves centrifuging the sample through a cut-off filter at an appropriate speed in order to efficiently filter the EVs according to size [66], starting with filters with pore sizes of 0.8 and 0.45 µm for the removal of larger particles and leaving a filtrate relatively enriched in EVs. Smaller vesicles are then eliminated from the filtrate using membranes with pores smaller than the desired EVs (0.22 and 0.1 µm) as a means of accessing the spent eluate (Figure 1). The size of the obtained EVs is defined by the range of maximum and minimum sizes of the first and last pore filtration membranes [67]. However, during filtration, the fluid is perpendicular to the membrane, which can cause target particle accumulation and filter clogging problems [68]. The tangential flow filtration (TFF) system is a filtration system where fluid passes parallel to the filter membrane rather than perpendicular to it, and when the TFF system separates impurities, such as extracellular proteins, the system combats the accumulation of target particles as a means of solving the Duchenne problem, thereby improving the purity and yield of EV production [69]. It has been reported that the use of TFF in conjunction with 3D improved cellular vesicle production by more than 2D-UC and 3D-UC, but conventional TFF systems are implemented as a single isolation unit with one membrane type, i.e., EV production [70]. Therefore, a separation method, namely double-cyclic TFF (dcTFF), has been proposed by researchers for more efficient separation of EVs in a specific nanometre range. Double-cyclic TFF has been found to significantly improve the efficiency of separation compared to filtration (DF) and single-cyclic TFF (scTFF) [68]. During the extraction and isolation of PDEVs, ultrafiltration is often combined with other methods for the extraction of PDEVs. For example, in combination with differential centrifugation, large-particle impurities are first removed by low-speed centrifugation, and then EVs are enriched by high-speed centrifugation. Finally, ultrafiltration is used for further purification to improve extraction efficiency and purity [71]. It can also be combined with size-exclusion chromatography. Different-sized EVs are separated based on differences in molecular size, and ultrafiltration assists in removing residual impurities and reducing the load and contamination of the chromatographic column [35].

#### 2.2.4. Size-Exclusion Chromatography (SEC)

Size-exclusion chromatography (SEC) is based on the separation of macromolecules on the basis of molecular size or hydrodynamic volume [72]. SEC can be used for separating EVs from a wide variety of sample matrices from prokaryotes and eukaryotes, including blood [73], follicular fluid, pomegranate, and milk [72]. Christian M et al. [54] used size-exclusion chromatography as a means of isolating EVs of pomegranate origin and found pomegranate juice to contain typically characterized EVs and that these extracellular vesicles had pharmacological properties, such as antioxidant and anti-inflammatory properties. Kaloyan Takov et al. [74] used ultracentrifugation and SEC to separate EVs in plasma, finding that EVs separated by SEC had higher purity and yield than those that were separated by ultracentrifugation. In addition, they discovered that SEC separation reduced protein contamination, so this method was applied to clinically relevant samples for proteomics analysis [75]. SEC has the advantages of shorter time consumption, high yield and recovery, and higher quality of isolated EVs used for protein and RNA diagnostics and in drug and drug delivery system studies [76]. EV isolation methods have advantages and disadvantages, and some researchers have found that the combined use of isolation methods improves the quality and yield of vesicles. For example, Nasibeh Karimi et al. [77] combined the use of SEC and sucrose density-gradient centrifugation as a means of increasing the purity of plasma-sourced EVs and reducing the contamination of lipoprotein particles. López de Las Hazas MC et al. [78] used a combination of ultracentrifugation and SEC for isolating and comparing EVs from four plants, namely broccoli, apple, orange, and pomegranate. They found that this significantly improved the purity of the EVs.

**Table 2 biology-14-00377-t002:** The principles, advantages, and disadvantages of extraction methods for PDEVs.

Extraction Method	Extraction Principle	Merit	Disadvantage	Reference
Ultracentrifugation (UC)	Separates vesicles by gradient centrifugation (such as differential centrifugation) combined with ultra-high speed centrifugation (>100,000× *g*)	Has a relatively high purity and can be used with multiple sample types.	Time-consuming, the equipment is expensive, the yield is low, and it may damage the vesicle structure.	[56]
Density gradient ultracentrifugation (DGU)	Separates vesicle and non-vesicle components using different density gradients.	Has higher purity and can distinguish vesicles of different densities	The operation is complex, the yield is low (with some vesicles being lost), and ultracentrifugation equipment is required.	[65]
Ultrafiltration (UF)	Utilizes filter membranes with different pore sizes to retain vesicles of specific sizes.	The operation is simple and rapid, and no expensive equipment is required.	Loss may occur due to the adsorption of the filter membrane, and the shear force may damage the vesicle structure.	[79]
Size-exclusion chromatography (SEC)	Separation according to the size of vesicles through porous media.	Retains the integrity of the vesicles and there is no damage caused by shear force.	Time-consuming, has a small processing capacity, and the equipment cost is high.	[80]

#### 2.2.5. The Stability of PDEVs

The stability and biological activity of PDEVs are closely related to the extraction methods. Before extraction and isolation, PDEVs are in a relatively stable internal environment within plants and are protected by structures, such as cell membranes, maintaining a stable state in terms of their components and structures. However, during the extraction and isolation processes, due to the variety of methods employed, the stability and biological activity of PDEVs can be affected to varying degrees. For example, conventional extraction methods, such as UC and UF, may damage the vesicle structure, thereby reducing their biological activity. Therefore, choosing an appropriate method during the extraction process is of utmost importance. Maintaining the stability of PDEVs is the key to preserving their biological activity. According to the research in “Minimal information for studies of extracellular vesicles (MISEV2023)” [81], the storage conditions before and after isolation can also have an impact on the stability and function of PDEVs. Kim K et al. [82] conducted research on Dendropanax morbifera leaf-derived extracellular vesicles (LEVs). The results showed that under storage conditions of −20 °C, 4 °C, 25 °C, and 45 °C, as the storage time and temperature increased, the particle size of LEVs gradually increased. Although it is currently recommended to store EVs in a low-temperature environment of −80 °C to maintain their stability for a certain period, repeated freezing and thawing can still have a negative impact on their stability and biological activity. After conducting cyclic freezing and thawing experiments on LEV, researchers found that the LEV after freezing and thawing had not only an increased particle size and changed shape but also a decreased cell uptake rate.

The stability of PDEVs can also be affected by different pH environments. In addition, the pH value may also have an effect on the size and surface charge of PDEVs. For example, when Jian-Hong Li [83] studied Houttuynia-derived exosome-like nanoparticles (HELNs), it was found that in a simulated gastric juice with a pH of 1.2 and a simulated intestinal juice with a pH of 6.8, when in an acidic environment (pH 1.2), the average particle size of HELNs increased to 212.4 nm and a decrease in negative charge occurred; while in an alkaline environment (pH 6.8), the average particle size increased to 203.7 nm and there was an increase in negative charge. Although changes occurred in the particle size and zeta potential, a certain degree of stability could still be maintained in different pH environments. Similarly, the research by Xiaozheng Ou et al. [84] on Catharanthus roseus (L.) Don leaves-derived exosome-like nanovesicles (CLDENs) also showed that in acidic and alkaline environments, the vesicle particle size increased, and this was more significant in the acidic environment. However, the vesicle membrane was not damaged and still maintained a membrane-vesicle-like structure.

The factors affecting the stability and biological activity of PDEVs are diverse. In addition to the extraction methods, storage temperature, and pH mentioned above, the different sources of PDEVs in plants, the reagents added during the extraction process, and the interactions with the container surface during storage may all have an impact on their stability and biological activity. An in-depth understanding of these factors is of great guiding significance for optimizing the extraction, isolation, and preservation methods of PDEVs, and will help to promote research and application development in related fields.

## 3. Biological Activities

### 3.1. Anti-Inflammatory Activity

A variety of factors cause inflammation, including dysregulation of the innate immune system [85], physical injury, ischemic injury, infection, and other types of trauma [86]. Accordingly, body inflammation can cause many diseases, including atherosclerosis [87], periodontal disease, cardiovascular disease, and systemic inflammation [88]. Inflammation occurs due to the induction or activation of biomarkers, such as tumor necrosis factor (TNF), interleukin-1 (IL-1), interleukin-6 (IL-6), interleukin-8 (IL-8), chemokines, cyclooxygenase, 5-lipoxygenase, and C-responsive proteins [89], or through the activation of signaling pathways, such as the transcription factors nuclear factor-κB (NF-κB), Nrf2/ARE [90], and Nrf2/HO-1 [91]. Generally, anti-inflammatory drugs, such as non-steroidal anti-inflammatory drugs (NSAIDs) [92], corticosteroids, and other anti-inflammatory drugs, can be used for inflammation. However, prolonged use of such drugs can lead to adverse reactions, such as anaphylactic, hematological, hepatic, and renal damage and gastrointestinal reactions [93]. PDEVs are considered to be potential biotherapeutic agents for various diseases due to their abundance of resources, high yield, low risk of immunogenicity in vivo, simplicity, safety, and low toxicity [94]. Mukesh K Sriwastva et al. [39] investigated the effect of mulberry bark-derived extracellular vesicles (MBELNs) against colitis and found that MBELNs prevent dextran sulphatesodium (DSS)-induced colitis in mice through the AhR/COPS8 pathway (Table 3). Stefania Raimondo et al. [45] found that lemon-derived extracellular vesicles (LEVs) inhibit macrophage TNF-α expression levels and increase the levels of NF-κB and ERK1-2 protein expression. It was also found that the anti-inflammatory mechanism of LEVs can be achieved by inhibiting the ERK/NF-κB signaling pathway. EVs extracted from Solanum nigrum were found to act as anti-inflammatory by decreasing the pro-inflammatory factor IL-6 against lipopolysaccharide-induced macrophage inflammation [38] and EVs secreted by garlic were also found to act as an anti-colitis treatment by inhibiting the secretion of pro-inflammatory factors, such as the biomarkers IL-6, IL-1β, TNF-α, and IFN-γ [33], which is in accordance with the results of a previous study. PDEVs, by virtue of their unique biological characteristics, play an indispensable role in the field of ant-inflammatory diseases. These nanoscale vesicles carry a variety of bioactive molecules, such as proteins, nucleic acids, and lipids. They can precisely regulate the intracellular signal transduction pathways through interactions with target cells, thus effectively suppressing the inflammatory response, promoting tissue repair and regeneration, and opening up new ideas and directions for treating inflammatory diseases.

### 3.2. Antitumor Activity

Cancer is considered a major health problem in modern society, with 19.3 million new cases and 10 million deaths predicted to occur worldwide during the 2020s. Female breast cancer is the most common cancer, followed by liver and lung cancer [95]. The most common cancer treatments include surgery, chemotherapy, radiotherapy, and immunosuppressant therapy, in which chemotherapeutic drugs can have significant side effects, including gastrointestinal reactions, hepatotoxicity, nephrotoxicity, and cardiotoxicity [96]. Therefore, to improve the quality of life with cancer, it is important to find a therapeutic drug that is effective and has fewer side effects. Studies have been conducted on the inhibition of cancer cell growth by plasma- and body-fluid-derived extracellular vesicles and there are solid studies of EVs of animal origin. Recently, PDEVs have also started attracting attention. For example, ginseng roots have several pharmacological properties, including anticancer, anti-inflammatory, antioxidant and anti-ageing properties. Therefore, investigating the inhibitory effects of ginseng root-derived nanoparticles (GDNPs) on melanoma revealed that GDNPs induced M1-like macrophage polarization through the toll-like receptor (TLR)-4/myeloid differentiation antigen 88 (MyD88) signaling pathway and enhanced total reactive oxygen species (ROS) production as a means of inducing apoptosis in mouse melanoma cells [42]. Lemon-derived extracellular vesicles (LDEVs) have been found to inhibit gastric cancer cells in vivo and in vitro, inducing S-phase arrest and apoptosis in the cell cycle of gastric cancer cells by increasing the levels of ROS [65]. Following the coincubation of tea-derived extracellular vesicles (TLNTs) with breast cancer cells for 5 h, over 80% of breast cancer cells were able to uptake TLNTs, thereby increasing reactive oxygen species (ROS) levels, leading to mitochondrial damage, cell-cycle arrest, and apoptosis of the tumor cells, which inhibit breast cancer cells [22]. Pancreatic cancer is one of the deadliest and least treatable malignancies, with a 5–10% 5-year survival rate, and therapeutic drugs are prone to the development of drug resistance. Research on the anticancer effects of broccoli-derived extracellular vesicles (BDEVs) on pancreatic cancer found that BDEVs can induce apoptosis in human pancreatic cancer cells by enhancing miR167a expression in the vesicles [50]. PDEVs have a potential future in cancer treatment and also play an important role in improving the efficacy of traditional cancer therapeutic drugs. For example, bitter gourd-derived extracellular vesicles (BMEVs) and 5-fluorouracil (5-FU) significantly ameliorate the resistance to chemotherapy-induced by the use of 5-FU alone when they are used concurrently in squamous cell carcinoma of the oral cavity [97]. PDEVs also act as efficient transporters in cancer therapy, and they can deliver exogenous proteins to human peripheral blood mononuclear cells and colon cancer cells, with significant uptake of exogenous proteins delivered by EVs in most organs in comparison to exogenous proteins delivered without EVs [98]. PDEVs, with their unique advantages, exhibit various potential roles in cancer treatment and provide new strategies and ideas for cancer therapy. Although there are still some problems and challenges at present, with the continuous in-depth research and the continuous progress of technology, these obstacles are expected to be overcome, promoting the application of PDEVs in clinical cancer treatment and bringing new hope to cancer patients.

### 3.3. Antioxidant Activity

Reactive oxygen species (ROS) are unstable molecules and include hydrogen peroxide (H_2_O_2_), hydroxyl radicals (-OH), monoclinic oxygen (^1^O_2_), and superoxide (O_2_^−^) [99]. Oxidative stress is when the body is subjected to external stimuli and produces too many ROS molecules, thereby exceeding its ability to remove them, which results in a dysregulation of the redox system. This then triggers oxidative damage, such as to intracellular proteins, and ultimately causes tissue and organ damage and apoptosis [100], which is generally associated with diseases including osteoporosis, atherosclerosis, chronic kidney disease, Alzheimer’s disease, inflammatory diseases, and diabetes [101,102,103,104,105,106]. When oxidative stress occurs in the body, this is an indicator that the production of free radicals in the body exceeds the ability of the body to scavenge them, so the body scavenges the excess free radicals to maintain the redox balance. Many natural, free radical scavengers, such as vitamin C (ascorbic acid), natural polysaccharides, and antioxidant dipeptides [52,107,108], play important roles in antioxidative stress. Citrus compounds, as one of the recognized natural sources of vitamin C, have always received much attention. Numerous studies have shown that organically grown fruits and vegetables not only contain no toxic substances but also have significantly higher extracellular vesicle contents than their conventionally grown counterparts. Moreover, the extracellular vesicles derived from organically grown fruits and vegetables exhibit a stronger antioxidant capacity, which is mainly attributed to their rich content of various antioxidants, such as vitamin C, catalase, glutathione, and superoxide dismutase [109,110]. Among them, vitamin C plays a key role in the treatment and prevention of leukeamia. High-dose vitamin C can induce the apoptosis of leukeamia cells. Based on this, Germana Castelli et al. [111] carried out research to explore whether EVs derived from organically grown grapefruits (ELPDNVs) have similar effects to vitamin C in the treatment of leukeamia. The research results show that ELPDNVs are rich in antioxidants, such as vitamin C, catalase, and glutathione, enabling ELPDNVs to inhibit the proliferation of leukeamia cells in a time-dependent manner. In contrast, the vitamin C sold on the market only has an anti-proliferative and cytotoxic effect on leukeamia cells at the highest dose (2 mM). Coincidentally, another study found that EVs derived from strawberries exhibit antioxidant effects in mesenchymal stromal cells, which is also closely related to their rich vitamin C content [112]. In the research process of EVs, in addition to the continuous exploration of EVs from single plant sources, researchers have recently found that composite PDEVs have more excellent antioxidant capabilities than single-PDEVs. This is mainly because composite PDEVs are composed of a mixture of EVs from multiple plants with more complex and diverse components, integrating the component advantages of multiple plants so as to play a synergistic role in various physiological activities. Recent research further shows that composite PDEVs play important roles in restoring the antioxidant–reduction balance [113] and promoting wound healing [114], and their mechanism of action is closely related to being rich in a series of antioxidants. Antioxidant enzyme is a type of enzyme that maintains the redox balance of the organism by neutralising ROS. Foreign antioxidant enzymes are rapidly removed when they are filtered through the glomerulus due to their short half-life, so packaging antioxidant enzymes in nanoparticles serves to protect the antioxidant enzymes from rapid removal and degradation [115]. As a result, the study of antioxidant nanoparticles has gradually emerged as a trend with the passage of time. Curcumin has demonstrated anti-inflammatory and antioxidant effects, but its use is limited due to low water solubility and bioavailability. Improving the water solubility and bioavailability of curcumin can improve its efficacy [116]. The encapsulation of drugs in nanoparticles can improve drug efficacy and reduce clearance, but the particle size, morphology, and surface topography of nanoparticles are all important parameters that determine the interaction of nanoparticles with biological organisms. Therefore, finding suitable synthetic processes for the drug encapsulation and purification of synthesised nanoparticles is more complicated [117]. Oxidative stress serves to induce the upregulation of the gene expression of the inflammation-associated factors IL-1β, IL-6, IL-8, and IL-12β, the activation of NF-κB, TLR-3, and TLR-7 transcript levels, and the downregulation of TGF-β1 gene expression [118]. Blueberry-derived exosome-like nanoparticles (BENVs) contain higher anthocyanin levels and inhibit the IL-8 protein production that is induced by hydrogen peroxide (H_2_O_2_)-induced oxidative stress in human colon cancer epithelial cells [119]. The biological activity of pomegranate-derived exosome-like nanoparticles (PgEVs) has been found to be consistent with the above experimental results, thereby inhibiting hydrogen peroxide (H_2_O_2_)-induced oxidative stress in human colon cancer epithelial cells without toxicity [54]. These experimental results demonstrate that PDEVs potentially play a therapeutic role in the fight against oxidative stress.

In the future, with continuous in-depth research, we expect to further identify the potential of PDEVs, opening up new paths for the treatment of related diseases and applications in the health field. For example, research may bring new breakthroughs and changes in the development of new antioxidant health products, the design of efficient skin repair products, and the exploration of innovative treatment plans for diseases, such as leukemia.

### 3.4. Anti-Infectious Activity

Infection is the process through which pathogens invade the body and interact with the organism, and there are bacterial, fungal, and viral infections. Bacterial infection refers to the disturbance of the normal functioning state of the organism following the invasion of bacterial pathogens into the human body, and Pseudomonas aeruginosa is a common Gram-negative bacterium, which is an important causative factor for serious human infections [120]. Pseudomonas aeruginosa is evident in diseases including bronchiectasis, meningitis and pneumonia [121,122,123] and is a bacterial pathogen that poses a threat to human health. There are also other bacterial pathogens that threaten human health, including Staphylococcus aureus, Escherichia coli, and Salmonella. Viral infection is when toxins are released by a virus that invades a host body and then reduces the immunity of said host body. Influenza A virus, Influenza B virus, respiratory syncytial virus, and neo coronavirus are all viral pathogens that can have a seriously negative impact on human health. Three major coronavirus outbreaks have been reported since 2002: SARS-CoV, MERS-CoV, and the recent 2019-nCoV (SARS-CoV-2) [124]. The novel coronavirus SARS-CoV-2 was named COVID-19 by the World Health Organisation (WHO) on 11 February 2020 and is one of the most common viruses that infect one of the highly pathogenic β-coronaviruses in humans [125], with preclinical symptoms of fever and cough progressing to pneumonia. Some bacteria, fungi, and viruses, like these that exist in nature, pose a serious threat to human health and anti-infective treatments have, therefore, become important. Antibiotics are the normal way to treat bacterial infections, but the resistance of bacterial pathogens to antibiotics seriously threatens the control of infectious diseases. The development of new antibiotics in recent years has slowed down due to cost and market profitability pressures, and there is now increased demand for new antimicrobial therapies [126]. After antiviral treatment for viral infections, some viruses develop a certain degree of resistance. Therefore, finding a new anti-infection treatment is urgent. Due to the widespread use of nanotechnology in recent years, researchers are increasingly interested in exosome-like nanoparticles. Qiao Zhuangzhuang et al. [26] used ginger-derived extracellular vesicle-like nanoparticles with Pd-Pt nanosheets as a means of fabricating EVs-Pd-Pt biomimetic nanoparticles. The study found EVs-Pd-Pt nanoparticles to have higher biocompatibility and long blood circulation, in addition to a good antimicrobial effect against Staphylococcus aureus in vitro and in vivo. In a study of the potential for tomato-derived extracellular vesicle-like nanoparticles to inhibit intestinal microbial pathogens, the nanoparticles were found to be more significantly able to inhibit Clostridium nucleatum and to have the ability to promote the growth of probiotic bacteria, such as Lactobacillus spp. [29]. In summary, PDEVs, with their natural characteristics and complex components, demonstrate great potential in the field of anti-infection. PDEVs can inhibit the infection processes of bacteria, viruses, and fungi through multiple mechanisms. They not only act directly on pathogens, interfering with the key links of their growth, replication, and invasion, but also effectively regulate the host immune response, activate immune cells, regulate the secretion of cytokines, and create an immune microenvironment conducive to resisting infection. Although current research on PDEVs is mostly at the basic experimental stage and there are challenges in large-scale preparation technology, analysis of in vivo action mechanisms, safety evaluation, and improvement of targeting, with the in-depth development of interdisciplinary research and continuous innovation and breakthroughs in biotechnology, these obstacles are expected to be gradually overcome. In the future, PDEVs are highly likely to become a powerful supplement or a completely new alternative to traditional anti-infection therapies, opening up new paths for the prevention and treatment of infectious diseases worldwide and significantly improving the health conditions and quality of life of patients.

**Table 3 biology-14-00377-t003:** In vivo studies on EVs derived from fruits and vegetables.

Plants	Research Object	Disease Model	Route of Administration	Results	Potential Mechanism	Reference
Ginger	C57BL/6j mice	Alcohol-induced liver damage	Oral administration	Reduced the levels of ALT, AST, and triglycerides	Inhibit the generation of ROS, activate theTLR4/TRIF pathway, and regulate the activity of Nrf2.	[127]
Lemon	BALB/c nude mice	Gastric cancer	Administration by injection	Reduced the tumor weight and inhibits the generation of ROS	Inhibit the generation of ROS and induce apoptosis of cancer cells	[46]
Ginseng	Balb/C mice and Wistar rats	Glioma	Wistar rats: IV Balb/Cmice: IC	Reduced the size of the tumor and decreased the luminescence intensity of C6 glioma	Reduce the expression of miRNA and chemokine genes related to cancer-associated fibroblasts (CAFs)	[128]
Tea	SD rats	IBS (irritable bowel syndrome)	Oral administration	Increased body weight, relieved defecation, and reduced hypersensitivity reaction	Regulate the CHR pathway to improve irritable bowel syndrome (IBS)	[129]
*Solanum lycopersicum*	C57BL/6J mice	Carotid artery restenosis injury	Oral administration	Reduced the neointimal area and the ratio of neointimal area to medial area, and attenuated the phenotypic transformation	miRNA164a/b-5p weakens phenotypic conversion and improves restenosis injury by activating the Keap1/Nrf2 pathway	[130]
Mulberry bark	C57BL/6J mice	Colitis	Oral administration	Increased body weight, inhibited colon shortening, and suppressed the release of inflammatory factors	Regulate the intestinal microbiota and activate the AhR—COPS8 pathway to improve colitis.	[39]
Goji	C57BL/6J mice	Muscle atrophy	Inject into the quadriceps femoris muscle	Increased grip strength, the cross-sectional area of the quadriceps femoris muscle, and the expression of myogenic regulatory factors	Improve muscle function by activating the AMPK/SIRT1/PGC1α pathway.	[131]
Orange	C57BL/6J mice	Obese (diet-induced)	Gavage	Restored intestinal function, reduced TG content, and regulate the immune response		[132]
Blueberry	C57BL/6J mice	Nonalcoholic fatty liver disease	Gavage	Reduced the mRNA levels of FAS and ACC1, as well as the contents of TC, TG, ALT, AST, and LDL-C, and increased the content of HDL-C	Antioxidative stress and inhibition of cell apoptosis.	[53]
Momordica charantia	C57BL/6J mice and Sprague Dawley rats	DOX cardiotoxicity	IV	Reduced cTnT and CK-MB and alleviated myocardial atrophy. Improved the new function indexes, such as EF, FS and HR	Activate the p62/Keap1/Nrf2 pathway to inhibit cell apoptosis.	[133]
Panax notoginseng	Sprague–Dawley rats	CI/R (cerebral ischemia–reperfusion injury)	IV	Reduced the area of cerebral infarction and inhibited the apoptosis of brain cells	Activate the PI3K/AKT signaling pathway to reduce the infarct area and improve cerebral ischemia–reperfusion (CI/R).	[37]

## 4. The Role of PDEVs as Drug Delivery Vehicles

Drug delivery systems deliver therapeutic drugs to the desired site, and these drug carriers can be natural or synthetic polymers. Drug delivery systems for synthetic drugs have been developed in recent decades to improve synthetic drug efficacy. However, these systems have limitations due to low immunogenicity, high cytotoxicity, and low efficiency [134]. At the same time, natural drug carrier systems have been developed, with the fastest growing and most prominent being EVs. EVs are important intercellular communication mediators and serve as drug carriers for treating inflammatory bowel disease [135], central nervous system disorders [136], rheumatoid arthritis [137], and other diseases, where EVs are derived from a variety of cells, body fluids, and plants. PDEVs are of significant interest as drug carriers, as they exhibit great potential as drug delivery nanocarriers because of advantages including efficient delivery, sustainability, low cost, low toxicity and low immunogenicity. Gingerol is a rich constituent of ginger that is classified as 6-Gingerol, 8-Gingerol, or 10-Gingerol on the basis of its structural formula and oil –water partition coefficient. The evaluation of the loading capacity of ginger-derived extracellular vesicles (GDEVs) loaded with three types of gingerols found that the amount of the three types of gingerols in the same mass of GDEVs was far higher than that of ginger slices [57]. Another study found that ginger-derived extracellular vesicles (GNs) coated on mesoporous silica nanoparticles (LMSNs) for making a composite nanoparticle significantly enhanced drug efficacy in oral administration [138]. In addition to ginger, there are other plants that can act as drug carriers. Sodium thiosulphate (STS) is a clinical drug that is used for vascular calcification (VC), but there are hindrances, including adverse effects and poor bioavailability. To solve these problems, researchers loaded sodium thiosulphate in EVs derived from grapefruit, finding it to exhibit highly effective therapeutic effects and low side effects when delivered by EVs of grapefruit origin [139]. Loading mRNA vaccines in EVs extracted from the juice of citrus helps protect the stability of the mRNA vaccine, which can be stored for up to a year at room temperature following lyophilization and encapsulation. This makes it an effective oral vaccine delivery strategy [140]. Extracellular vesicle drug carrier systems of mammalian origin have been studied intensively and are widely recognised as having low immunogenicity, biodegradability, and low toxicity. The EV drug carrier system of plant origin not only possesses the characteristics of mammalian EVs but also has the advantages of high yield and is completely natural. Compared with synthetic drug carriers, it has higher biocompatibility, lower immunogenicity, and a greater likelihood of crossing the blood–brain barrier (BBB) of various organisms [141]. Therefore, there is great potential for PDEV drug–carrier systems to become a therapeutic approach in the future.

## 5. Conclusions

Plant-derived extracellular vesicles (PDEVs), as emerging carriers of bioactive substances, play a crucial role in communication between cells and between cells and the environment. They have demonstrated great potential in the biomedical field. Research achievements in their extraction methods and applications in antitumor, anti-inflammatory, anti-infective, and antioxidative stress aspects have provided new ideas and directions for the future development of medicine, but they are also accompanied by many challenges. Currently, the production and purification of PDEVs face significant challenges. In the production process, not only are large amounts of raw materials required, but energy consumption and equipment costs are also extremely high. In the extraction and separation process, common methods, such as ultracentrifugation and ultrafiltration have exposed many drawbacks when applied on a large scale. For example, ultracentrifugation relies on professional equipment, has a very limited processing capacity, cannot meet the needs of large-scale industrial production, is relatively inefficient, has high costs, and is also prone to damaging the vesicles during operation, leading to a decrease in their activity. The ultrafiltration method also has similar problems and struggles to meet the efficiency and cost requirements of large-scale production. To solve these problems, the industry has tried to combine methods, such as ultracentrifugation and ultrafiltration, with electrophoresis and dialysis, which has increased the yield and reduced the cost to a certain extent. However, new problems have emerged. The products obtained by the combined methods have a relatively low purity, which affects the further application of PDEVs [56]. If a technical path combined with highly efficient purification methods can be explored in the future to achieve high-efficiency, low-cost, and high-purity production, the scalability of PDEVs in the industrial field will be full of hope and is expected to bring new development opportunities to related industries. More and more studies have found that PDEVs have potential effects in anti-inflammatory, antitumor, antioxidative stress, and anti-infective aspects (Figure 2). From the antitumor perspective, PDEVs show unique advantages. They can carry a variety of bioactive molecules, such as proteins, nucleic acids, and small-molecule metabolites, and exert an antitumor effect by regulating the signal pathways related to tumor cell proliferation, apoptosis, and metastasis. Some studies have shown that PDEVs can deliver specific microRNAs into tumor cells, interfering with the expression of oncogenes and, thus, inhibiting the growth of tumor cells [142]. However, the application of PDEVs in tumor treatment still faces challenges, such as how to achieve the targeted delivery of PDEVs, improve their enrichment efficiency in tumor tissues, and deeply understand their long-term safety and potential side effects. In the fields of anti-inflammation and anti-infection, PDEVs also show good application prospects. PDEVs can regulate the activity of immune cells, inhibit the release of inflammatory factors, and alleviate the inflammatory response [143]. In terms of anti-infection, the antimicrobial peptides, plant hormones, and other substances carried by them can directly inhibit the growth of pathogens or activate the host’s immune defense mechanism. However, a more in-depth study is needed on the mechanism of action of PDEVs in complex inflammatory and infectious microenvironments to optimize their treatment strategies. In terms of oxidative stress-related diseases, the antioxidative stress function of PDEVs has also attracted much attention. The antioxidants rich in PDEVs, such as vitamin C, glutathione, and superoxide dismutase, can scavenge excessive reactive oxygen species in the body and protect cells from oxidative damage [109]. Although some studies have reported that the antioxidant effect of composite plant-derived extracellular vesicles is stronger than that of single plant-derived extracellular vesicles, a large number of basic and clinical studies still need to be carried out on how to further improve the antioxidant capacity of PDEVs and effectively apply them to the treatment of clinical oxidative stress-related diseases.

Overall, PDEVs show promising prospects in multiple disease treatment fields. However, from basic research to clinical application, many obstacles still need to be overcome. In the future, it will be necessary to integrate multidisciplinary knowledge, optimize extraction techniques, deeply explore the mechanism of action, and solve problems, such as delivery and safety, to promote the translational application of PDEVs in the biomedical field.

## Figures and Tables

**Figure 1 biology-14-00377-f001:**
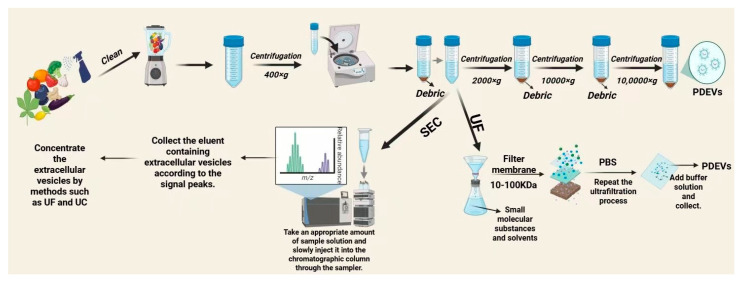
Methods for extracting PDEVs.

**Figure 2 biology-14-00377-f002:**
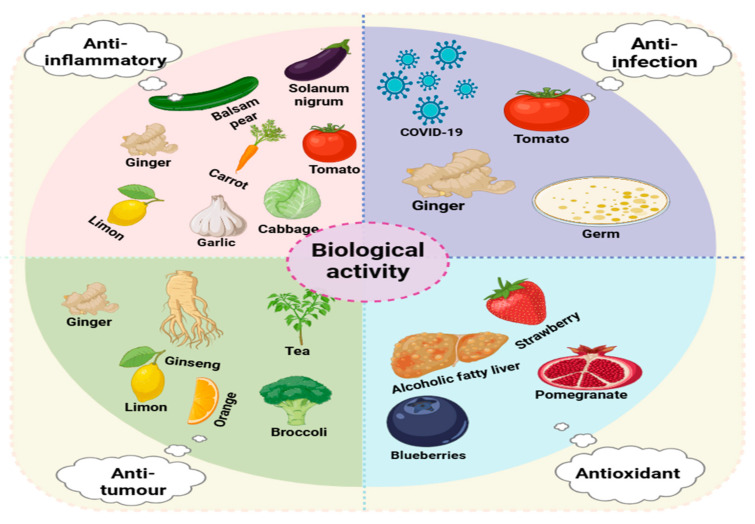
Biological activities of plant-derived extracellular vesicles (PDEVs).

**Table 1 biology-14-00377-t001:** Extraction methods, characterization, and biological activities of extracellular vesicles derived from fruits and vegetables.

Plants	Source	Extraction Techniques	Biological Function	Illnesses	Average Particle Size (nm)	References
Ginger	Vegetable	UC	Anti-inflammatory, anti-infection, antitumor	COVID-19	70.09 ± 19.24	[25,26,27]
Tomato	Vegetable	UC	Anti-inflammatory, anti-infection	Inflammatory-related diseases	110 ± 10	[28,29,30]
Cabbage	Vegetable	UF, SEC	Anti-inflammatory	Inflammatory-related diseases	100	[31]
Momordica charantia	Vegetable	Density gradient centrifugation	Anti-inflammatory	Colitis	106.0	[32]
Garlic	Vegetable	UC, density gradient centrifugation	Anti-inflammatory, liver protection	Colitis	43.82–396.1	[33,34]
Carrot	Vegetable	SEC, UF	Anti-inflammatory		143.9	[35,36]
Panax notoginseng	Root	UC, density gradient centrifugation	Anti-inflammatory	Cerebral ischemia–reperfusion injury	151.3	[37]
*Solanum nigrum* L.	Vegetable	PEG	Anti-inflammatory	Inflammatory-related diseases	107.0	[38]
Mulberry bark	Bark	UC	Anti-inflammatory	colitis	151.3 ± 45.4	[39]
Grapefruit	Fruit	Density gradient centrifugation	Antitumor	Melanoma	210.8 ± 48.62	[40,41]
Ginseng	Vegetable	Density gradient centrifugation	Antitumor, regenerative	Melanoma	92.04 ± 4.85	[42,43,44]
Lemon	Fruit	UC	Antitumor, anti-inflammatory	Gastric cancer, chronic inflammation	65 ± 2.7	[45,46,47]
Orange	Fruit	UC	Antitumor	Ovarian cancer	91	[48,49]
Tea	Leaf	UC, gradient centrifugation	Antitumor	Breast cancer	166.9	[22]
Broccoli	Vegetable	UC, SEC	Antitumor	Pancreatic cancer	146.7 ± 7.2	[50,51]
Strawberry	Fruit	UC	Antioxidant	Inflammatory-related diseases	30–191	[52]
Blueberry	Fruit	UC	Antioxidant	Alcoholic fatty liver disease	189.62	[53]
Pomegranate	Fruit	SEC	Antioxidant	Inflammatory-related diseases	148.7 ± 9.2	[54]

## Data Availability

Not applicable.

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
