# Peer review of "Advances in Plant-Derived Extracellular Vesicle Extraction Methods and Pharmacological Effects"

_biology, 2025, doi:10.3390/biology14040377_

Round 1
Reviewer 1 Report
Comments and Suggestions for Authors
Advances in Plant-Derived Extracellular Vesicles Extraction Methods and Pharmacological Effects
By Nuerbiye Nueraihemaiti , et al
Of course a review on all the potential application of Plant-derived extracellular vesicles (PDEVs) are welcome.
This review mostly deals with methods to obtain PDEVs and in this sense it is fairly complete, while literature is increasing day by day. What is not treated is the issue on which is the most suitable approach to obtain industrial scalability. This is very important inasmuch as for instance the approach to obtain PDEVs from cultures of vegetables’ derived cells does not allow a real industrial scalability, together with including growth factors and antibiotics in the PDEVs’ purification. Moreover, the issue on the source of fruits and vegetables to obtain PDEVs is entirely disregarded. This last is a key point inasmuch as in general EVs belong to a framework aimed at scavenging toxic or unwanted material; in turn meaning that PDEVs from intensive farming may contain pesticides and microbicides, that all the methodologies to obtain PDEVs by the way concentrate transforming a potential highly healthy product in a poison. Another issue is the use of PDEVs’ mixes in order to provide a product containing different anti-oxidants (Logozzi M, Di Raimo R, Mizzoni D, Fais S. Nanovesicles from Organic Agriculture-Derived Fruits and Vegetables: Characterization and Functional Antioxidant Content. Int J Mol Sci. 2021 Jul 29;22(15):8170. doi: 10.3390/ijms22158170. PMID: 34360936; PMCID: PMC8347793.).
Lastly, the authors did not include discussion on recent papers showing that PDEVs (i) may cure leukemia ( Castelli G, Logozzi M, Mizzoni D, Di Raimo R, Cerio A, Dolo V, Pasquini L, Screnci M, Ottone T, Testa U, Fais S, Pelosi E. Ex Vivo Anti-Leukemic Effect of Exosome-like Grapefruit-Derived Nanovesicles from Organic Farming-The Potential Role of Ascorbic Acid. Int J Mol Sci. 2023 Oct 27;24(21):15663. doi: 10.3390/ijms242115663. PMID: 37958646; PMCID: PMC10648274. ); (ii) may revert heavy redox imbalance ( Di Raimo R, Mizzoni D, Spada M, Dolo V, Fais S, Logozzi M. Oral Treatment with Plant-Derived Exosomes Restores Redox Balance in H2O2-Treated Mice. Antioxidants (Basel). 2023 May 29;12(6):1169. doi: 10.3390/antiox12061169. PMID: 37371899; PMCID: PMC10295262. ) ; (iii) may cure skin lesions ( Di Raimo R, Mizzoni D, Aloi A, Pietrangelo G, Dolo V, Poppa G, Fais S, Logozzi M. Antioxidant Effect of a Plant-Derived Extracellular Vesicles' Mix on Human Skin Fibroblasts: Induction of a Reparative Process. Antioxidants (Basel). 2024 Nov 9;13(11):1373. doi: 10.3390/antiox13111373. PMID: 39594515; PMCID: PMC11590891. )
The authors should at least read the published reviews below:
Orefice NS, Di Raimo R, Mizzoni D, Logozzi M, Fais S. Purposing plant-derived exosomes-like nanovesicles for drug delivery: patents and literature review. Expert Opin Ther Pat. 2023 Feb;33(2):89-100. doi: 10.1080/13543776.2023.2195093. Epub 2023 Mar 27. PMID: 36947052.
Logozzi M, Di Raimo R, Mizzoni D, Fais S. The Potentiality of Plant-Derived Nanovesicles in Human Health-A Comparison with Human Exosomes and Artificial Nanoparticles. Int J Mol Sci. 2022 Apr 28;23(9):4919. doi: 10.3390/ijms23094919. PMID: 35563310; PMCID: PMC9101147.
Comments on the Quality of English Languageit needs only an overview from a mother tongue person
Author Response
Comments 1: What is not treated is the issue on which is the most suitable approach to obtain industrial scalability.
Response 1: Thank you very much for your detailed and insightful comments on our manuscript. We fully acknowledge the significance of determining the most suitable approach for achieving industrial scale-up of plant-derived extracellular vesicles (PDEVs). In the discussion section of the revised manuscript, we have conducted an extensive literature review. The goal is to identify potentially promising methods for large-scale PDEVs production. For instance, we intend to surmount the limitations of current methods by integrating them with alternative techniques. This integration is expected to enhance the extraction rate, thereby enabling the realization of industrial scalability.
Comments 2: Moreover, the issue on the source of fruits and vegetables to obtain PDEVs is entirely disregarded.
Response 2: We are extremely grateful for the valuable suggestions you put forward. We have rectified the plant sources in Table 1. Thank you for your suggestions. Once again, we sincerely apologize for any inconvenience caused to you.
Comments 3: Another issue is the use of PDEVs’ mixes in order to provide a product containing different anti-oxidants (Logozzi M, Di Raimo R, Mizzoni D, Fais S. Nanovesicles from Organic Agriculture-Derived Fruits and Vegetables: Characterization and Functional Antioxidant Content. Int J Mol Sci. 2021 Jul 29;22(15):8170. doi: 10.3390/ijms22158170. PMID: 34360936; PMCID: PMC8347793.).
Response 3: We are extremely grateful for the valuable suggestions you put forward. We have added content related to this literature in the "Antioxidant Stress" section and made citations, which enriches the discussion. (References 110)
Comments 4: Lastly, the authors did not include discussion on recent papers showing that PDEVs (i) may cure leukemia ( Castelli G, Logozzi M, Mizzoni D, Di Raimo R, Cerio A, Dolo V, Pasquini L, Screnci M, Ottone T, Testa U, Fais S, Pelosi E. Ex Vivo Anti-Leukemic Effect of Exosome-like Grapefruit-Derived Nanovesicles from Organic Farming-The Potential Role of Ascorbic Acid. Int J Mol Sci. 2023 Oct 27;24(21):15663. doi: 10.3390/ijms242115663. PMID: 37958646; PMCID: PMC10648274. ); (ii) may revert heavy redox imbalance ( Di Raimo R, Mizzoni D, Spada M, Dolo V, Fais S, Logozzi M. Oral Treatment with Plant-Derived Exosomes Restores Redox Balance in H2O2-Treated Mice. Antioxidants (Basel). 2023 May 29;12(6):1169. doi: 10.3390/antiox12061169. PMID: 37371899; PMCID: PMC10295262. ) ; (iii) may cure skin lesions ( Di Raimo R, Mizzoni D, Aloi A, Pietrangelo G, Dolo V, Poppa G, Fais S, Logozzi M. Antioxidant Effect of a Plant-Derived Extracellular Vesicles' Mix on Human Skin Fibroblasts: Induction of a Reparative Process. Antioxidants (Basel). 2024 Nov 9;13(11):1373. doi: 10.3390/antiox13111373. PMID: 39594515; PMCID: PMC11590891. )
Response 4: We are extremely grateful for the valuable suggestions you put forward. During the revision process, we have made reasonable citations and in-depth discussions of the references you mentioned in section 3.3, Antioxidant Activity of section 3, Biological Activity. The integration of these references has greatly enriched the content of the article and significantly improved its quality. We deeply regret that our oversight has caused inconvenience to your review work.
Comments 5: The authors should at least read the published reviews below:
Orefice NS, Di Raimo R, Mizzoni D, Logozzi M, Fais S. Purposing plant-derived exosomes-like nanovesicles for drug delivery: patents and literature review. Expert Opin Ther Pat. 2023 Feb;33(2):89-100. doi: 10.1080/13543776.2023.2195093. Epub 2023 Mar 27. PMID: 36947052.
Logozzi M, Di Raimo R, Mizzoni D, Fais S. The Potentiality of Plant-Derived Nanovesicles in Human Health-A Comparison with Human Exosomes and Artificial Nanoparticles. Int J Mol Sci. 2022 Apr 28;23(9):4919. doi: 10.3390/ijms23094919. PMID: 35563310; PMCID: PMC9101147.
Response 5: We are extremely grateful for the valuable suggestions you put forward. We have added content related to this literature in the "Antioxidant Stress" section and made citations, which enriches the discussion. (References 109)
Comments on the Quality of English Language
Comments 1:it needs only an overview from a mother tongue person
Response 1: We are extremely grateful for the valuable suggestions you put forward. We have polished the main body of the text.
Thank you again for being able to consider our manuscript for publication in biology.
Professor Wenting Zhou
Department of Pharmacology
School of Pharmacy Xinjiang Medical University, Xinjiang, China.

Reviewer 2 Report
Comments and Suggestions for Authors
1- The whole manuscript has to be reexamined for Grammatical errors.
2-Introduction—second line—not microcapsule, its microvesicles, and please refer to actual articles that referred to these names.
3- "Three subtypes of EVs have been isolated so far in Anopheles . "I think it's Arabidopsis. 4- "These nanovesicles generally have a small size range of between 10 nm and 300 nm"- Reference?
5- My major concern about this review is that there's no mention of MISEV ( minimal information for studies on extracellular vesicles); they have laid out all the dos and Don'ts when we start EV research. The authors should include MISEV 2023 references.
6- The authors mentioned the different methodologies used to isolate EVs, but none of the references match any published plant EV papers. Authors have to include them, give proper references.
7- Table -1 is beneficial.
8- Figure 3: Can you color code the 4 categories?
Comments on the Quality of English Language
Overall, it's a good collection of literature. As a review, please add relevant references. Also, please go through the whole manuscript and correct the grammar. There are many grammatical errors in it. Be on the lookout for spelling mistakes. A few sentences come out as plagiarism; rewriting those sentences would be ideal. The introduction section lacks the historical perspective of EV research, and It would be great to emphasize the first plant EV paper and its techniques. Citing another review paper is not a good idea, and this review has many such instances. Please cite the original paper and not the review.
Author Response
Thank you for giving us an opportunity to revise our manuscript, and we truly appreciate reviewer for their valuable suggestions and comments on our manuscript entitled “Advances in Plant-Derived Extracellular Vesicles Extraction Methods and Pharmacological Effects” (Manuscript ID: biology-3515131), which we wish to be considered for publication in biology.
Comments 1: The whole manuscript has to be reexamined for Grammatical errors.
Response 1: We are extremely grateful for the valuable suggestions you put forward. We have carefully checked and corrected the grammatical errors, and also carried out text polishing. Therefore, we sincerely apologize for any inconvenience caused to your reading.
Comments 2: Introduction—second line—not microcapsule, its microvesicles, and please refer to actual articles that referred to these names.
Response 2: We are extremely grateful for the valuable suggestions you put forward. We have made revisions to the manuscript by strictly referring to the relevant content of the references.
Comments 3: "Three subtypes of EVs have been isolated so far in Anopheles . "I think it's Arabidopsis.
Response 3: We are extremely grateful for the valuable suggestions you put forward. We have revised the manuscript according to the original references.
Comments 4: "These nanovesicles generally have a small size range of between 10 nm and 300 nm"- Reference?
Response 4: We are extremely grateful for the valuable suggestions you put forward. We have revised the manuscript in accordance with the original references and provided of the references.
Comments 5: My major concern about this review is that there's no mention of MISEV ( minimal information for studies on extracellular vesicles); they have laid out all the dos and Don'ts when we start EV research. The authors should include MISEV 2023 references.
Response 5: We are extremely grateful for the valuable suggestions you put forward. We have carefully read this reference and cited it in the part about the stability of plant - derived extracellular vesicles in the main text.(References 81)
Comments 6: The authors mentioned the different methodologies used to isolate EVs, but none of the references match any published plant EV papers. Authors have to include them, give proper references.
Response 6: We have carefully examined the main text and accurately cited the references related to the extraction of plant - derived extracellular vesicles in the extraction method section. We deeply regret the inconvenience caused to you during your reading due to this oversight.
Comments 7: Table -1 is beneficial
Response 7: Thank you for your recognition of this table. We have adjusted the order of the plants in the table according to the sequence of the content in the article.
Comments 8: Figure 3: Can you color code the 4 categories?
Response 8: We are extremely grateful for the valuable suggestions you put forward. We have carried out color classification for Figure 3.
Thank you again for being able to consider our manuscript for publication in biology.
Professor Wenting Zhou
Department of Pharmacology
School of Pharmacy Xinjiang Medical University, Xinjiang, China.

Reviewer 3 Report
Comments and Suggestions for Authors
- The title is clear and aligns with the journal.
- Missing the objective, it is not clear after the introduction.
- These are additional gaps needed in this review:
- Biological activity and stability of PDEVs before and after extraction and separation methods.
- Clinical application of PDEVs, it is a lack of comprehensive studies on their pharmacological effects, including their mechanisms of action, efficacy, and safety profiles in vivo.
- Information about the efficiency of PDEV extraction methods, ensuring the intact bilayer structure and bioactivity during extraction and storage.
- The bibliography is adequate.
- The figures and tables are appropriate, but you need to mention this at the text and reorder section 3 Biological activities
- Add a table with in vivo studies of PDEVs.
- The document does not include future perspectives, particularly in the conclusion section.
- The English must be improved (Consistency in terminology, sentence structure, grammar and punctuation, avoid repetition).
- Incorporate diagrams and a flowchart showing the extraction process of PDEVs or a table comparing the advantages and advantages of different extraction methods.
- Provide mor detailed explanations of technical terms and processes.

- The English must be improved (Consistency in terminology, sentence structure, grammar and punctuation, avoid repetition).
Author Response
Thank you for giving us an opportunity to revise our manuscript, and we truly appreciate reviewer for their valuable suggestions and comments on our manuscript entitled “Advances in Plant-Derived Extracellular Vesicles Extraction Methods and Pharmacological Effects” (Manuscript ID: biology-3515131), which we wish to be considered for publication in biology.
Comments 1: The title is clear and aligns with the journal.
Response 1: Thank you very much for your comments.
Comments 2: Missing the objective, it is not clear after the introduction.
Response 2: We are extremely grateful for the valuable suggestions you put forward. We have supplemented the objectives in the introduction section. We sincerely apologize for the inconvenience caused to your reading.
Comments 3: Biological activity and stability of PDEVs before and after extraction and separation methods.
Response 3: We are extremely grateful for the valuable suggestions you put forward. We have supplemented the content regarding the stability of PDEVs in the article and discussed the stability and biological activities of PDEVs before and after extraction.
Comments 4: Clinical application of PDEVs, it is a lack of comprehensive studies on their pharmacological effects, including their mechanisms of action, efficacy, and safety profiles in vivo.
Response 4: We are extremely grateful for the valuable suggestions you put forward. They are of great help in improving the quality of our article. We have added Table 3 on the mechanisms and pharmacological effects of in-vivo experiments and also made relevant supplements in the main text.
Comments 5: Information about the efficiency of PDEV extraction methods, ensuring the intact bilayer structure and bioactivity during extraction and storage.
Response 5: We are extremely grateful for the valuable suggestions you put forward. We have supplemented Table 2 to illustrate the efficiency of extraction methods and the impact of these methods on the integrity of the bilayer structure of PDEVs. In addition, in the section on the stability of PDEVs, we have explored the effects of different storage conditions on their biological activity and stability.
Comments 6: The figures and tables are appropriate, but you need to mention this at the text and reorder section 3 Biological activities.
Response 6: We are extremely grateful for the valuable suggestions you put forward. We have mentioned Table 1 at the corresponding positions in the main text and rearranged the contents of Table 1 according to the order of biological activities. This ensures that the order of biological activities presented in Table 1 is completely consistent with the order elaborated in Section 3, aiming to enhance the relevance and logicality between the article content and the chart.
Comments 7: Add a table with in vivo studies of PDEVs.
Response 7: We are extremely grateful for the valuable suggestions you put forward. We have added a table on in-vivo research and presented it in the form of Table 3 to present the relevant content more intuitively and clearly.
Comments 8: The document does not include future perspectives, particularly in the conclusion section.
Response 8: We are extremely grateful for the valuable suggestions you put forward. We have rewritten the discussion section, further optimizing the analysis and elaboration. Meanwhile, we have added the content of prospects prospects, providing forward-looking insights into the subsequent development directions of the research, so as to make the article more complete and profound.
Comments 9: The English must be improved (Consistency in terminology, sentence structure, grammar and punctuation, avoid repetition).
Response 9: We are extremely grateful for the valuable suggestions you put forward. We have polished the main body of the text.
Comments 10: Incorporate diagrams and a flowchart showing the extraction process of PDEVs or a table comparing the advantages and advantages of different extraction methods.
Response 10: We are extremely grateful for the valuable suggestions you put forward. To present the extraction principles, advantages, and disadvantages of various extraction methods more intuitively, we have added Table 2, which enables readers to compare and understand them at a glance.
Comments 11: Provide mor detailed explanations of technical terms and processes.
Response 11: We are extremely grateful for the valuable suggestions you put forward. In Fig 2, we have presented in detail the specific procedures of various methods for extracting extracellular vesicles from plant sources.
Thank you again for being able to consider our manuscript for publication in biology.
Professor Wenting Zhou
Department of Pharmacology
School of Pharmacy Xinjiang Medical University, Xinjiang, China.

Round 2
Reviewer 1 Report
Comments and Suggestions for Authors
The authors have addressed my comments.
Reviewer 3 Report
Comments and Suggestions for Authors
In Table 1, include a note below the table explaining the abbreviations (UC, UF, SEC).
Below Table 1, there is an incomplete number "2" and the numbers "3" and "4." Please remove them, as they appear to be typographical errors.
In Section 2.2.5, the label "2.2.4" appears. Please verify and correct it if necessary.
In the title of Section 2, decide whether to use "Characterisation" or "Characterization" for consistency.
In Section 2.2.2, use the abbreviation "DGU" in both the text and Table 1.
Ensure that any abbreviation introduced at the beginning of a paragraph is consistently used throughout the paragraph.
In Table 2 and 4 , add lines to separate the methods for better readability, and use the corresponding abbreviations in the paragraphs.
Verify where Figure 3 and Tab 3 is mentioned in the text.
In Table 3, include a note below the table explaining the abbreviations used.
Comments on the Quality of English LanguageEnglish must be revised and improved